# La(OH)_3_ Multi-Walled Carbon Nanotube/Carbon Paste-Based Sensing Approach for the Detection of Uric Acid—A Product of Environmentally Stressed Cells

**DOI:** 10.3390/bios12090705

**Published:** 2022-09-01

**Authors:** Sara Knežević, Miloš Ognjanović, Vesna Stanković, Milena Zlatanova, Andrijana Nešić, Marija Gavrović-Jankulović, Dalibor Stanković

**Affiliations:** 1Faculty of Chemistry, University of Belgrade, Studentski trg 12-16, 11000 Belgrade, Serbia; 2University of Bordeaux, CNRS, Bordeaux INP, Institut des Sciences Moléculaires, UMR 5255, 33400 Talence, France; 3Department of Theoretical Physics and Condensed Matter Physics, “VINČA” Institute of Nuclear Sciences-National Institute of the Republic of Serbia, University of Belgrade, 11000 Belgrade, Serbia; 4Scientific Institution, Institute of Chemistry, Technology and Metallurgy, National Institute University of Belgrade, 11000 Belgrade, Serbia; 5Department of Radioisotopes, “VINČA” Institute of Nuclear Sciences-National Institute of the Republic of Serbia, University of Belgrade, 11000 Belgrade, Serbia

**Keywords:** La(OH)_3_@MWCNT, electrochemical sensor, uric acid, DAMP molecule, cell damage, stress

## Abstract

This paper aims to develop an amperometric, non-enzymatic sensor for detecting and quantifying UA as an alert signal induced by allergens with protease activity in human cell lines (HEK293 and HeLa). Uric acid (UA) has been classified as a damage-associated molecular pattern (DAMP) molecule that serves a physiological purpose inside the cell, while outside the cell it can be an indicator of cell damage. Cell damage or stress can be caused by different health problems or by environmental irritants, such as allergens. We can act and prevent the events that generate stress by determining the extent to which cells are under stress. Amperometric calibration measurements were performed with a carbon paste electrode modified with La(OH)_3_@MWCNT, at the potential of 0.3 V. The calibration curve was constructed in a linear operating range from 0.67 μM to 121 μM UA. The proposed sensor displayed good reproducibility with an RSD of 3.65% calculated for five subsequent measurements, and a low detection limit of 64.28 nM, determined using the 3 S/m method. Interference studies and the real sample analysis of allergen-treated cell lines proved that the proposed sensing platform possesses excellent sensitivity, reproducibility, and stability. Therefore, it can potentially be used to evaluate stress factors in medical research and clinical practice.

## 1. Introduction

Various events can strain the human body, putting stress on the cells and tissues. Oxidative stress, heat shock, hypersensitivity, autoimmune diseases and other disorders can cause cell dysfunction and lead to apoptosis [1]. Besides passive release, dying cells produce an increased amount of uric acid (UA), which can indicate both the environmental and medical conditions that induce cell damage [2]. DAMPs are molecules that have a physiological role inside the cell but acquire additional functions when released from the cells: they alert the body about danger, stimulate an inflammatory response, and promote the regeneration process [3]. Apart from their passive release by dead cells, some DAMPs can be secreted or exposed by living cells undergoing life-threatening stress.

Uric acid, being a final product of purine metabolism, has a high impact on human health. It is produced in the liver, intestines, kidneys, vascular endothelium, and muscles by the metabolism of purine [2].

Both high and low human plasma UA concentrations can indicate a pathological state. On the one hand, UA acts as a scavenger of peroxyl radicals, hydroxyl radicals, and singlet oxygen, thus exhibiting antioxidative features. It is a specific inhibitor of radicals generated by the decomposition of peroxynitrite, preventing cell injuries and nitration of tyrosine residues in proteins. Furthermore, it can protect against oxidative damage by chelating metal ions such as iron and copper, lowering their catalytic activity in free-radical reactions [4]. UA also shows neuroprotective properties and reduces the risk of multiple sclerosis, Parkinson’s disease, Alzheimer’s disease, and optic neuritis [5].

UA also acts as an initiator and amplifier of allergic inflammation and can cause hypertension and cardiovascular diseases via the induction of growth factors, hormones, cytokines, and autacoids. UA penetrates vascular smooth muscle fibers and activates signal transduction, increasing the expression of inflammatory mediators. Furthermore, urate crystals can deposit in the connective tissues of the joints, tendons, kidneys, and rarely in heart valves and the pericardium. Consequently, UA is a risk factor for renal disorders (kidney injuries and kidney stones), acute and chronic inflammatory arthritis, gout, myocardial infarction, and stroke [2,5]. Elevated serum uric acid levels are associated with insulin resistance and diabetes mellitus [2].

The important UA feature is that it is produced in higher concentrations when the cells suffer from stress. Both live and dying cells degrade their nucleic acids in deamination and dephosphorylation processes. Purine nucleoside phosphorylase converts the relevant degradation products, inosine and guanosine, to the purine bases, hypoxanthine and guanine, which further metabolize to xanthine. Oxidation of xanthine via xanthine oxidase leads to the formation of UA. Therefore, although uric acid is regularly present in cells, it increases in concentration when the cells are damaged [2]. The UA’s release from dying cells can serve as an indicator of unfavorable environmental factors or pathological states.

Considering its importance and abundance in the human body, it is no wonder that numerous methods have been developed for UA detection and determination. The separation and detection methods commonly involve chromatography or electrophoresis coupled with UV/VIS [6,7,8,9,10] spectroscopy or electrochemical detection (voltammetry [11,12,13,14,15,16], ECL [17,18,19,20], and amperometry [21,22,23,24,25]), including both enzymatic and nonenzymatic approaches [26]. Recently, attention has shifted toward electrochemical sensors and biosensors because they enable the fast, direct, and precise determination of the analyte in the complex biological matrix.

Enzyme-based UA biosensors use uricase as an enzyme for the oxidation of uric acid [27,28]. The enzyme-based approach is expensive, and the obtained sensor is sensitive to environmental conditions, lacks reproducibility, and requires complicated immobilization of the enzyme. Non-expensive and robust non-enzymatic sensors perform direct oxidation of UA on the surface of the electrode material [29,30]. Various materials, such as covalent organic and metal incorporated conductive polymers [13,16,19,23], metal oxides [30,31,32], carbon-based materials, and their composites [11,12,33], have been tested for UA detection [5,26]. These nanomaterials have been recognized as promising materials for the development of analytical methods for the detection not only of UA but also of numerous other biologically active compounds [34,35,36,37].

The use of La [32,38,39]-based electrode materials has been reported in the literature with the main applications in fuel cells [40,41,42] and sensing devices [43,44,45,46], while Wang et al. [47] used LaFeO**_3_** for the simultaneous determination of dopamine, uric acid and ascorbic acid, thus proving the materials’ compatibility with these analytes. La doping can affect lattice structures and phase transformations of compounds due to its larger atomic radius and unique electron structure, thus enhancing the catalytic and electrochemical properties of La-doped materials [32]. Furthermore, Guo et al. [48] suggested that the alkaline properties of La(OH)**_3_** contribute to the acidic analytes’ bonding. To improve the conductivity and increase the active surface and electron transfer of the electrode, lanthanum hydroxide was incorporated into the composite with multi-walled carbon nanotubes (MWCNTs), which possess exceptional mechanical and physicochemical properties [38,49].

This study aims to develop an electrochemical sensor for the fast, accurate, and precise measurement of UA released from damaged cells, or rather for monitoring the dependence between environmental factors and the stress they cause to the cells/tissues (Figure 1). An electrochemical sensor was developed using a glassy carbon paste electrode modified with newly synthesized nanomaterial consisting of MWCNTs decorated with La(OH)**_3_**. The obtained sensor was used for the amperometric detection of uric acid under optimal experimental conditions. Furthermore, the influence of common interfering substances on UA determination was examined, as well as the reproducibility, repeatability, and stability of the proposed sensor. We showed that the proposed method could be used to evaluate stress factors in medical research and clinical practice.

## 2. Materials and Methods

### 2.1. Cell Cultures

Cell lines cultivated for the allergen treatment included HEK293 and HeLa. HEK293 cells were cultivated in Dulbecco’s modified Eagle medium (DMEM, Sigma-Aldrich, St. Louis, MO, USA) supplemented with 10% heat-inactivated fetal bovine serum (FBS), 1% pen/strep (penicillin 10,000 U/mL; streptomycin 10 mg/mL) and 200 mM L-glutamine (Sigma-Aldrich, St. Louis, MO, USA), and HeLa cells were cultivated in Eagle’s minimum essential medium (EMEM, Lonza, Basel, Switzerland) supplemented with 20% FBS, 1% pen/strep (penicillin 10,000 U/mL; streptomycin 10 mg/mL) and 1% (*v/v*) 200 mM L-glutamine. Cell lines were grown in a humidified atmosphere of 95% air and 6% CO_2_ at 37 °C, in T-25 flasks (Thermo Scientific, Waltham, MA, USA) until confluence and then were trypsinized (0.25% Trypsin-0.53 mM EDTA). HEK293 (250,000 cells per mL) and HeLa (300,000 cells per mL) cells were seeded at a volume of 1 mL in 12-well plates (Sarstedt, Nümbrecht, Germany) and grown to confluence before allergen treatment.

### 2.2. Cysteine Protease Treatment

Two proteolytic enzymes, papain from Carica papaya and actinidin from Actinidia deliciosa, were employed for the induction of acute cellular response in the HEK293 and HeLa cell lines. Commercial papain from papaya was purchased from Sigma–Aldrich (P 4762, Sigma–Aldrich, St. Louis, MO, USA), while actinidin was isolated from kiwifruit (Actinidia deliciosa) according to the previously published procedure [50]. Before the treatment, actinidin (0.9 mg/mL) was incubated for 1 h at 37 °C in DMEM (enzyme activation) or DMEM with an equimolar amount of E64 inhibitor (enzyme inactivation). Before the treatment, papain (0.4 mg/mL) was dissolved in the medium without or with an equimolar amount of E-64 cysteine protease inhibitor. Confluent HEK293 and HeLa cell monolayers were treated with papain, inactivated papain, activated actinidin, inactivated actinidin, or DMEM/EMEM without FBS at 37 °C for 0 (1 h before the treatment), 3, 6, or 12 h, respectively. The leakage of UA from HEK293 and HeLa cells into the medium was measured using the electrochemical sensor La(OH)**_3_**@MWCNT/CP.

### 2.3. Reagents and Apparatus

The crystal structure of La(OH)**_3_**@MWCNT nanocomposite was examined using powder X-ray diffraction (XRD) data collected using a high-resolution SmartLab^®^ X-ray diffractometer (Rigaku, Japan) with a Cu Kα radiation source. The operating current and voltage were 30 mA and 40 kV, respectively. For the experiments, the dried powders were flattened with a zero-background silicon plate. Diffraction data were collected in the range of 10–60° 2θ with a step of 0.03° and recording speed of 2°/min. The average crystallite size was estimated by applying Scherrer’s equation on the most intensive diffraction peaks (Ls = κλ/βscos(θB. The morphology and surface properties of nanocomposite used for electrode modification were investigated using a field emission-scanning electron microscope FE-SEM MIRA3 (Tescan, Czech Republic) operating at 20 keV. The magnification of the composites was in the range of 10–100,000 times. The samples were prepared by fixating the conductive tape on a holder, vacuum drying, and spray-coating with gold using a Sputter coater.

All electrochemical measurements were carried out using a CH Instruments analyzer (Austin, TX, USA) driven by voltammetric software CHI (Version 4.03). A three-electrode system was employed, with a modified carbon paste (CP) working electrode (WE), a Calomel reference electrode (RE), and a Pt wire as the counter electrode (CE). The measurements were conducted in 10 mL of 0.1 phosphate buffer (PB) at pH 6. Electrochemical characterization of working electrodes was conducted using electrochemical impedance spectroscopy (EIS) and cyclic voltammetry (CV) in 5 mM K**_3_**[Fe(CN)**_6_**]/K**_4_**[Fe(CN)**_6_**] (1:1) in 0.1 M KCl solution. EIS measurements were conducted in the frequency range from 10 kHz to 10 mHz, at 0.3 V. The used experimental parameters for CV measurements were: potential range from −0.5 V to 1 V and the scan rate of 50 mV/s. The CV measurements were also performed in the 0.1 M PB (pH 6) solution containing 10 µM UA Chronoamperometry (CA) was the detection and quantification method of choice. Measurements were taken in 0.1 M PB at pH 6, with the potential set on 0.3 V while adding an increasing quantity of UA during the period of 600 s. High-performance liquid chromatography (HPLC) for uric acid analysis was conducted (Dionex Ultimate 3000, Thermo Fisher, Waltham, Massachusetts, USA) with photodiode array detection on a Hypersil Gold aQ C18 analytical column (150 mm × 3 mm, 3 μm).

### 2.4. Material Preparation

#### 2.4.1. Synthesis of La(OH)_3_@MWCNT Nanocomposite

La(OH)**_3_**@MWCNT powder was prepared by means of the method reported in the literature [51], after some modifications. At the very beginning, a 1 M solution of LaCl**_3_** was prepared by dissolving the proper amount of corresponding salt in ultra-pure water. A volume (5 mL) of the starting solution was treated with vigorous and constant stirring. Then, the same volume of 2 M K**_2_**CO**_3_** solution (5 mL) was rapidly introduced into this mixture, with the addition of a few drops of 1 M NaOH solution to initiate precipitation. After the precipitation was complete, the precipitate was filtered and washed with distilled water several times until the washed water became neutral (pH = 7). Furthermore, the obtained product was washed three times with ethanol and then dried at 100 °C for 6 h.

#### 2.4.2. Electrode Preparation

An unmodified glassy carbon paste electrode was prepared by mixing 80% of glassy carbon powder and 20% of paraffin oil in a mortar to form the homogeneous carbon paste. Modified glassy carbon paste electrodes were made by mixing 1.6 weight percent of La(OH)**_3_**, 0.4% of MWCNTs and 2, 5, and 10% of La(OH)**_3_**@MWCNTs (with the La(OH)**_3_** to MWCNT balance being 4:1) in the unmodified paste. Subsequently, the working electrode was filled with the paste, and the electrode surface was polished with paper, washed with DI water, and directly used for measurements.

## 3. Results and Discussion

### 3.1. Characterization

The XRD patterns of synthesized La(OH)_3_@MWCNT are shown in Figure 1A. It is clear that the sample is highly crystalline and the diffraction peaks at 15.6°, 26.3°, 27.9°, 31.5°, 36.0°, 39.5°, 47.1°, 48.7°, 49.9°, and 55.3° match well with the crystal planes of (100), (110), (101), (200), (111), (201), (002), (211), (102), (112). All of the reflections can be assigned to lanthanum hydroxide having a hexagonal symmetry and P63/m space group (JCPDS #36-148128) [52]. The average crystallite size of La(OH)_3_ nanoparticles is (26 ± 4) nm, calculated by Scherrer’s equation. The peaks from other phases occurring at 19.6° and 42.3° can be assigned to (002) and (110) reflections of graphitic MWCNT [53]. This can be proof of successful composite formation.

The morphology and the shape of prepared La(OH)_3_, as well as the La(OH)_3_@MWCNT composite, are analyzed by FE-SEM. Different magnifications of the composite are displayed in Figure 1B,C. The La(OH)_3_ nanoparticles are scattered over long multi-walled carbon nanotubes. This makes the surface of the nanotubes significantly larger, which can potentially affect the electrochemical properties of the composite. The rice-like lanthanum hydroxide nanoparticles are partially agglomerated, with an average diameter between 100 and 200 nm, and a width ranging from 30 to 60 nm (inset of Figure 1C).

### 3.2. Treatment of HEK293 and HeLa Cells with Cysteine Proteases

Many allergens have biological activities, including enzymatic ones. Exposure to proteases via the respiratory tract can induce the release of UA into the airway lumen, and promote type 2 immune response [53]. In this study, two allergens with cysteine protease activity, papain and actinidin, were employed as stressor agents for the induction of UA release as an endogenous danger signal molecule. Papain is a potent proteolytic enzyme [54] (EC 3.4.22.2) and allergen from papaya. Actinidin is a major allergen from kiwifruit (Actinidia deliciosa), which was purified under native conditions. The purified enzyme preserved cysteine protease activity [54] and was employed for cell treatment. Detection of UA was performed in the respective cell culture medium after the treatment of HEK293 cells as well as HeLa cells with papain and actinidin after 0, 6, 12, and 24 h, respectively. Papain was a potent inducer of UA release from both cell lines in a time-dependent manner, while actinidin did not induce detectable amounts of the UA molecule. Although the molecular mechanisms for UA release upon allergen treatment are not clarified in detail, papain exhibits five times higher proteolytic (caseinolytic) activity than actinidin [54].

### 3.3. Electrocatalytic Properties of the Electrode Materials

Characterization of the electrodes’ surfaces during the optimization of the final electrode modification was performed using CV and EIS. CV was employed to determine the interfacial properties of electrode materials and examine electron transfer kinetics between the electrode surface and the electrolyte. Firstly, the unmodified carbon paste electrode was compared to La(OH)_3_-, MWCNT-, and La(OH)_3_@MWCNT-modified CP electrodes to define the optimal nanomaterial composition to use for electrode modification. CVs were performed in PB at pH 6 in the 5 mM redox probe K_3_[Fe(CN)_6_]/K_4_[Fe(CN)_6_] and 0.1 M KCl support electrolyte, at the scan rate of 50 mV/s (Appendix A). The oxidation and reduction peaks that originate from the Fe^2+/3+^ redox pair are visible on every voltammogram obtained using unmodified and modified CP electrodes (Appendix A). The oxidation peak currents, for 2% (weight percent) of individual modifiers in the paste amount 16.73 μA, 12.39 μA, 16.87 μA and 18.21 μA for CP, La(OH)_3_/CP, MWCNTs/CP and La(OH)_3_@MWCNTs/CP, respectively. Consequently, La(OH)_3_@MWCNTs/CP was determined to have the best electrocatalytic properties and was chosen for further examination. The next step in the electrode’s surface optimization was varying the modifier percentage in CP. The resulting voltammograms (Appendix A) show a steady rise in oxidation peak current with the increase in the share of the modifier in CP. The peak current values equaled 18.13 μA for 2%, 23.47 μA for 5%, and 30.81 μA for 10% of La(OH)_3_@MWCNTs in the CP. Furthermore, with the increase in the modifier content in CP from 2% to 10%, the oxidation peak potential shifted from 0.46 V to 0.32 V, with peak-to-peak separations (ΔE) = 0.69 V, 0.47 V and 0.36 V and oxidation/reduction current ratios (Ia/Ic) = 1.02, 1.01 and 1.04 for 2%, 5% and 10% La(OH)_3_@MWCNTs/CP, respectively. Higher values for the peak-to-peak separation in this system, in comparison with the theoretical value of 59 mV, is a common phenomenon and it is the product of the heterogeneity of the hand-made carbon paste electrodes. The reported decrease in the ΔE value, as a result of the increase in the amount of the modifier, indicate excellent properties of the selected composite regarding the diffusion properties of the electrode surface. Since oxidation occurs on lower potentials and the other parameters point out that the electrode reaction is reversible when using 10% La(OH)_3_@MWCNTs/CP, this electrode material is assumed to have the best electrocatalytic properties.

To further support the abovementioned assumptions, effective surface areas were estimated for each electrode. Calculations were performed using the Randles–Sevcik Equation and the specific surface areas of the electrodes were 2.02 mm^2^, 1.50 mm^2^, 2.04 mm^2^, 2.20 mm^2^, 2.83 mm^2^ and 3.72 mm^2^ for CP, La(OH)_3_/CP, MWCNT/CP, 2% La(OH)_3_@MWCNT/CP, 5% La(OH)_3_@MWCNT/CP and 10% La(OH)_3_@MWCNT/CP, respectively. This proves, once again, that composite material formation is a crucial part of the modification process. Not only does it enhance the material’s electrocatalytic properties, but also leads to an increase in the effective surface area.

In addition to estimating the electron transfer resistance of the chosen electrode and comparing its values to those obtained using different materials, EIS gives information on other conductivity/resistance-related properties of the electrode system, such as double-layer capacitance or diffusion rate. EIS spectra consist of a semicircle (high frequency) and a linear (low frequency) region. The semicircle radius is electron transport resistance-dependent and is defined by its Rct value, while the linear part is diffusion-dependent. The measurements were conducted in PB pH 6, in the 5 mM redox probe containing Fe^2+/3+^ redox pair from cyano complexes and 0.1 M KCl support electrolyte. Rct values of CP, La(OH)_3_/CP, MWCNTs/CP and La(OH)_3_@MWCNTs/CP (Appendix A) were 40,480 Ω, 29,956 Ω, 48,345 Ω and 35,160 Ω, respectively.

Contrary to our beliefs, these values indicate that MWCNTs have the poorest electrocatalytic features. However, La(OH)_3_ encourages the electron shuttle and thus has the lowest Rct value, while the composite material exhibits improved properties when compared to the unmodified paste and MWCNTs alone. Moreover, as previously discussed, this material shows the best current response in the cyclic voltammetry measurements, which confirms that La(OH)_3_@MWCNTs exhibits the optimal electrocatalytic activity. Furthermore, the increase in the La(OH)_3_@MWCNTs amount in the paste leads to the reduction in Rct and thus the improvement of the overall electrochemical performance, which is evident from the experimental data (Appendix A). The obtained Rct values are 35,015 Ω, 29,889 Ω, and 19,559 Ω for the 2%, 5%, and 10% La(OH)_3_@MWCNTs/CP, respectively. In short, better CV response and the decrease in the Rct values with the increase in the modifier content prove the positive impact of the composite formation and the synergetic effect of its components on the electron transfer kinetics enhancement.

### 3.4. The Material Behavior in the Presence of UA

To further examine electrodes’ compatibility with the chosen analyte, their electrochemical response was measured in the presence of UA. The measurements were conducted in 10 mM uric acid solution in 0.1 M PB pH 6, in the potential range from 0.2 V to 1 V and at the scan rate of 50 mV/s. Cyclic voltammograms of CP, La(OH)_3_/CP, MWCNTs/CP, and La(OH)_3_@MWCNTs/CP electrodes show the oxidation peak current values 0.31 µA, 0.28 µA, 0.27 µA, and 0.27 µA, respectively (Appendix A). Considering only anode peak height, we could assume that electrode modification does not significantly influence the electrodes’ performance in the presence of UA. We could also incorrectly conclude that the unmodified CP electrode shows the best properties. However, reflecting on the complete potential area, it is obvious this is not the case. We can see that with the electrode modification, the resolution of the peak improves and residual current decreases. This leads to better peak to peak separation and to a lowering of the detection limit. Furthermore, while the oxidation of the analyte occurs on the same potential on every voltammogram, the electrolysis of the supporting electrolyte shifts to higher potentials as we proceed with the electrode modification, from 0.49 V for CP to 0.67 V for La(OH)_3_@MWCNTs/CP. This additionally lowers capacitive current, improves peak shape, and enables a stable and reproductive environment for detecting low analyte concentrations.

When compared to an unmodified CP electrode, modified electrodes (MWCNTs/CP, La(OH)_3_/CP, and La(OH)_3_@MWCNT/CP) provide a significantly improved voltammetric peak and higher currents (I) toward UA detection. The CV signals of UA using these modified electrodes were similar, so for the selection of the optimal electrode for further experiments, we used the data obtained from the electrochemical characterization of all electrodes, as previously described. The La(OH)_3_/CP electrode had the poorest CV response based on these measurements, most likely due to a decrease in electrode active surface area. MWCNTs promoted electron transfer and decreased the interfacial resistance, and better CV response was achieved when MWCNTs/CP were used. The La(OH)_3_ likely makes the entire complex of MWCNTs and La(OH)_3_ more porous, which can help improve the response of the modified electrode. All of this led to the conclusion that the synergistic effect of MWCNTs and La(OH)_3_ improved the modified electrode’s electrochemical properties, which will contribute to the sensor’s analytical performance for uric acid detection. The electrochemical behavior of La(OH)_3_@MWCNTs/CP electrodes, with different weight percent of the modifier (from 2% to 10%) was compared using CV (Appendix A). It is evident from the graph that the material manifests a catalytic effect on the electron shuttle, with the oxidation peak currents being 0.27 µA, 0.36 µA, and 0.45 µA for the electrodes containing 2%, 5%, and 10% of the modifier. Furthermore, the increase in the material contained in the paste was followed by oxidation peak potential shift to lower values, from 0.39 V for 2% to 0.33 V for 10% of the composite material in CP, in addition to a further decrease in the capacitive current. Therefore, the material containing 10% of the composite was determined to be the most suitable for further development of the sensing platform for UA detection in human cells. Additionally, we recorded the CV of three different concentrations of UA, using the electrode with 10% of the composite material in CP (Appendix A).

### 3.5. Optimization of pH of the Supporting Electrolyte. Study of the Reaction Kinetics-Influence of Varying Scan Rates on the Material

For optimization of the pH of the supporting electrolyte, CV measurements were performed with La(OH)_3_@MWCNTs/CP electrode containing 10% of the modifier mixed in the carbon paste. Measurements were taken in 0.1 M PB, in the pH range from 2 to 9, at the scan rate of 50 mV/s, and they confirm that the reaction is pH-sensitive. The more alkaline the solution becomes, the oxidation peak potential shifts to lower values, from 0.56 V at pH 2 to 0.24 V at pH 9 (Figure 2A). On the contrary, peak current values increase with the rise in pH, only to gradually drop afterwards, reaching the maximum of 0.45 µA at pH 6. Furthermore, peak resolution is fairly good on lower pH values, ending with pH 6 where the peak is well defined and narrow, unlike the broad peaks at higher pH. The obtained results correlate well with UA’s pKa [55] values and electrocatalytic oxidation mechanism (Figure 2) [55].

The reported mechanism can be divided into three main steps—2e-/2H^+^ deprotonation and the oxidation of uric acid, the hydration of the intermediate diimine and its subsequent decomposition into allantoin and CO_2_. While this mechanism infers that the reaction is pH-dependent and that alkaline conditions allow the initial deprotonation to proceed, it is less obvious as to why the pH values higher than pH 6 are less favorable. Since pKa values for uric acid are 5.4 and 9.8, we can assume that the elevation of pH above pH 6 leads to further deprotonation of urate or diimine, thus introducing new intermediate species and broadening the oxidation peak, lowering its resolution and decreasing the maximal peak current.

The best performance of the electrochemical system at pH 6, and its similarity with the physiological conditions are the reasons why this pH is used for all further measurements.

The CV measurements in PB pH 6 were conducted with optimized electrode surface parameters at various scan rates (from 2 mV/s to 100 mV/s) to determine the reaction kinetics. The oxidation peak potential and ΔE are constant for all scan rates in the measured range (Figure 2B), while the peak current values (inset Figure 2B) increase linearly with the increase in the scan rate, which is described by the equation I(nA) = 3.2436 v (mV/s) + 0.5213, with the linear regression coefficient R = 0.9921. This means that the UA solution is stable, the electron transfer processes are fast, and the electrode reaction is an adsorption-controlled process. From the equation i_p = (n^2^ F^2^)/4RT vAΓ *, we can calculate the surface coverage of the adsorbed species (Γ*) [56]. The surface coverage values for the scan rates from 2 mV/s to 100 mV/s are 0.11 µmol/cm^2^, 44.08 nmol/cm^2^, 22.04 nmol/cm^2^, 11.02 nmol/cm^2^, 7.35 nmol/cm^2^, 5.51 nmol/cm^2^, 4.41 nmol/cm^2^, 2.94 nmol/cm^2^ and 2.20 nmol/cm^2^. These values indicate that, although the adsorption of the UA on the electrode surface decreases with the increase in the scan rate, it is not negligible and can influence the accuracy of the measurements. To prevent introducing a systematic error in the measurements, the surface of the electrode was renewed before each measurement (the excess paste was squeezed out of the electrode, polished with a clean piece of paper and washed with deionized water). The simple and reproducible restoration of the clean electrode surface was one of the major advantages of using the carbon paste electrode in this study.

Even though the anodic peak does not show potential shift and its current linearly increases with the scan rate, the reduction peak is almost absent in all voltammograms, while the cathodic capacitive current increases with the increase in the scan rate. This indicates that the electrode reaction is characterized by a reversible electron transfer followed by an irreversible chemical reaction [56]. In this case, the irreversible chemical reaction is allantoin formation, and it is incorporated in the electrochemical step.

### 3.6. Uric Acid Detection

Amperometric determination was used for UA quantification due to its low detection limit, rapidness, wide linear range, easy signal processing, and the possibility of following the real-time concentration changes in flow systems. The amperometric response of the La(OH)_3_@MWCNTs/CP electrode, containing 10% of nanomaterial mixed in the CP, toward different standard solutions of the UA, was recorded in 0.1 M PB pH 6, at the fixed potential at 0.3 V (Figure 3A). Successive UA addition to the solution was accompanied by a corresponding current increase and the calibration plot I (nA) against c (µM) was constructed (Figure 3B). The linear response was in the range from 0.67 μM to 121 μM UA with the detection limit, calculated from the plot as 3 S/m (where S is the standard deviation of the blanc and m is the slope of the calibration plot), 64.28 nM, and the analogously calculated limit of quantification (10 S/m) was 0.22 μM. The calibration curve follows the trend expressed by the linear regression equation I (nA) = 2,1582 + 0.4430 c (μM), with the linear regression coefficient R = 0.9969.

Five successive measurements of 2 μM UA standard solution were performed, to test the repeatability of the proposed sensing platform for UA determination and monitoring, and the obtained current values were 6.01 nA, 5.71 nA, 5.70 nA, 5.71 nA and 5.42 nA, giving an RSD of 3.65% (Appendix A). The reproducibility was estimated with five different electrodes, which were constructed independently using the proposed procedure (Appendix A). The RSD is 4.13% for the peak current measured in 2 μM UA in PBS (pH 6.0), which demonstrates the reliability of the fabrication procedure. The stability of the modified electrode was also studied using CV. When the electrode was cyclically swept for 30 cycles, the decrease in the initial responses of the modified electrode was 3.6%.

Furthermore, we conducted an extensive literature overview and compared the linear working range, LOD and RSD of our method to the best methods documented in the literature (Table 1). The results suggest that the proposed sensing platform is comparable, if not better than those previously reported. Several UA detection methods using only carbon nanotubes as the electrode material have been reported [21,57]. Although these sensors have adequate linear ranges, they generally have slightly higher detection limits compared to electrode utilizing modified materials, or require complex purification techniques prior analysis. By using MWCNTs decorated with La(OH)_3_ nanoparticles to modify the CP electrode, a very good sensitivity of the UA detection method was achieved. This is probably due to the joint action of the modifier—the alkaline properties of La(OH)_3_, which facilitate the binding of UA, as well as the increase in the active surface area and electron transfer efficiency by MWCNTs.

### 3.7. Interference Studies

One of the essential features of the sensor is its selectivity. CV measurements of a 10 µM UA solution in 0.1 M PB at pH 6 were conducted in the presence of common interferents found in biological matrices. Both organic and inorganic interfering species which exist in the lysed cell culture were taken into account. The measurements are performed in the potential range from −0.5 V to 1 V, at the scan rate of 50 mV/s, with the ratio of UA and potential organic interferents 1:1. The chemical behavior of UA in the presence of ascorbic acid (AA), citric acid (CA), dopamine (DOPA), gallic acid (GA) and glucose (GLU) is shown in the Appendix A. While CA and GLU do not influence UA detection, the other interferents lead to signal elevation. However, the peak originating from UA remains visible and measurable in the presence of DOPA, AA and GA (Appendix A). Although considerable interferences originating from the substances commonly present in the biological matrix present the major drawback of the proposed method, they can be overcome by careful experiment planning and execution and/or the use of chemometric methods. We proved this by measuring only the height of the peak originating from UA oxidation, which remains nearly the same in all samples, with the highest deviation for the ascorbic acid (histogram on Appendix A).

Considering the fact that high concentrations of salts can be present in the sample, the influence of 0.1 M KCl, NaCl, CaCl_2_, MgOAc and NaOAc in 0.1 M PB at pH 6 on detection of 10 mM UA was studied. The results show that none of the mentioned salts in a 1000:1 ratio to UA influence UA detection. The interference study proves that the proposed sensing platform shows satisfying selectivity towards UA in the presence of common organic and inorganic interferents.

The practicability estimation of the developed method after the cell stress.

To analyse the applicability of the proposed sensing platform in biological samples, measurements were conducted in a real sample matrix. CV measurements were performed in the potential range from −0.5 V to 1 V at the scan rate of 50 mV/s in the presence and absence of UA to scan for potential interferents (Appendix A). The baseline in the biological matrix gives a high current response, originating from the oxidation of electrochemically active species contained in the matrix. However, this does not influence the UA amperometric detection, because, at 0.3 V, where uric acid oxidation occurs, no matrix interferences are present. To prove this, a calibration curve is constructed once again in the biological sample matrix (DMEM). The obtained results are displayed by the amperometric curve (Appendix A) and corresponding plot (inset Appendix A). The amperometric current response changes linearly with the UA concentration in the range from 1 µM to 38 µM, which is described by the equation I (nA) = 4.1455 + 0.4326c µM, with the linear regression coefficient R = 0.9910.

After the calibration, real sample analysis was conducted. The samples were obtained by treating the human HEK293 and HeLa cell cultures with the papain, and the actinidin allergen, respectively, during a time frame of 12 h. Ten microliters of the sample was added to 5 mL of 0.1 PB at pH 6, and the change in the amperometric current at 0.3 V was measured. However, the UA concentration in the sample was found to be lower than the limit of quantification. Therefore, the sample was spiked with UA standard solution, so the final concentration of UA (spiked) was 100 µM. The measurements were repeated five times with the spiked sample, and the mean value of the rise in the amperometric signal corresponded to the 0.2010 mM UA concentration in the 5 mL of PB, which agrees with the total UA concentration in the sample of 100.5 µM. That proves that the proposed method is applicable in real sample analysis. However, it requires either more concentrated samples, larger quantities of samples, or spiking the samples with a known amount of UA.

To further support this claim, a series of HeLa and Hek293 cell cultures were treated with papain and EG4 inhibited papain for prolonged periods. The cell cultures were prepared in duplicate with each of the abovementioned allergens independently and the amperometric response at 0.3 V was recorded immediately after the addition of allergens, and after incubation periods of 3 h, 6 h and 12 h. The obtained results are given as mean values of the measurements in Table 2. The control groups of cells, without the added allergens, were also analyzed to compare the UA release from the treated and the untreated cells.

The acquired values indicate that the stress on the cells after the adequately long (6 h and 12 h) exposure to papain is sufficient to release considerable amounts of UA. Furthermore, the quantity of UA released is directly proportional to the incubation period, proving that the developed method is suitable for real-time detection or monitoring of cell damage or stress.

For method validation, the same real samples were analyzed by using HPLC as a standard method. The obtained results display good agreement with those achieved in the developed electrochemical method. The recoveries of the determination are 94.3–103.0%, indicating good accuracy of the developed method in real sample analysis.

### 3.8. Environmental Impact of the Analysis

While developing an analytical procedure, attention is being shifted more and more to its ecological acceptability. Several methods can assess the green aspects of an analytical procedure in which the green analytical procedure index (GAPI) (Figure 3A) and Analytical GREEnness metric approach and software (AGREE) (Figure 3B) are used. While GPAI is the most comprehensive and has the advantages of graphically describing all relevant factors-quantification; sample collection, preservation, transport and storage; sample preparation; reagents and compounds used and instrumentation, AGREE enables straightforward interpretation, since the level of greenness is expressed numerically on a scale from 0 to 1, with 1 being the greenest. Furthermore, AGREE contains all the relevant data about the analytical procedure and is easy to generate thanks to a free software downloadable from https://mostwiedzy.pl/AGREE (accessed on 7 June 2022) [59]. For GAPI, each aspect is presented with a pentagon further divided into segments that stand for specific demands that make a procedure green (amounts of solutions used, purification, energy consumption, generated waste, etc.). The colors of the segments depict the low, medium or high impact of the method on the environment, moving from green through yellow to red, and each of them has to fulfil certain conditions to be considered green [60].

In Figure 3, we can see that the analysis performed with the proposed sensing platform is quantitative and can be performed at-line. Thus, no sample preservation, transport or storage is needed. Furthermore, the cell culture matrix can be used directly or dissolved in water or biological buffer without any prior purification or sample preparation. This minimizes the energy consumption and the amount of generated waste, while the minimum volume needed for the analysis is 5 mL. Overall, this analytical procedure can be considered green, direct and robust and it can easily be further miniaturized for commercial application.

## 4. Conclusions

This paper developed and tested a sensing platform for UA determination in the biological sample. Knowing that the electrode surface modifications are the key enabler for next-generation chemistries based on the interface reactions, we offered a sensing platform based on the novel composite material, optimized to have the best electrochemical performance. Their compatibility with the UA as the analyte was tested and the calibration was performed in the optimal conditions, showing remarkable sensitivity and a wide dynamic working range. Reproducibility and repeatability tests have shown excellent accuracy and precision of the method as a limiting and key factor for practical application. The disadvantage of this method is its limited selectivity in the presence of the common interfering species in the biological matrix. However, this drawback can be bypassed by good experiment planning and/or the use of chemometrics. In the end, the proposed sensing platform was tested in the human cell cultures exposed to the allergens. In the first series of samples (stressed with the papain, and actinidin allergen), concentrations of UA were too low to perform the direct measurement, so the samples had to be spiked. In spite of that, we were able to determine the UA concentration and to repeat the measurements with the unspiked samples stressed with papain. In this way, we not only proved that the stress induced in the cells can be measured by this method, but clearly distinguished the levels of stress induced in HeLa and Hek 293 cell cultures by the use of three different allergens (papain, and actinidin). Since UA is released when cells suffer from stress, sensing devices that can measure a change in the UA concentration can provide us with plenty of information on both the environmental factors and pathological states that cause stress to the human organism. Furthermore, the simplicity of this method and the device itself opens up opportunities for its commercial use. Hence, we think that this sensing device, as it is, can potentially find applications in medical research and clinical practice.

## Data Availability

The study did not report any data.

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
