# Peer review of "La(OH)3 Multi-Walled Carbon Nanotube/Carbon Paste-Based Sensing Approach for the Detection of Uric Acid—A Product of Environmentally Stressed Cells"

_biosensors, 2022, doi:10.3390/bios12090705_

Round 1

Reviewer 1 Report

This paper presented a non-enzymatic UA sensor based on carbon paste La(OH)3@MWCNT modified electrode, for detecting of environmentally stressed damage of cells. The study is interesting and useful. However, Revision is still needed to improve this paper for publication.

1.   Because UA could be electrochemically oxidized on CNT directly, the role of La(OH)3 should be declared more clearly.

2.   Please provide the CV curves of the electrode tested with different UA concentrations.

3.   The table 1 and table 2 are not well presented. Please provide tables with grid line, and aligned contents.

4.  In figure 3, Calibration curve need to add the error bar.

5. Some recent publications might be cited, such as https://doi.org/10.1016/j.jelechem.2020.114888.

Reviewer 2 Report

This manuscript reports an electrochemical sensor for rapid and sensitive detection of uric acid (UA) based on using La(OH)3/multi-walled carbon nanotube (MWCNT)/carbon paste (CP) complex as the sensing surface. The authors demonstrated that the sensor allowed for the detection of uric acid in a linear range of 0.67-1221 µM with a detection limit of 64.28 nM. The sensor was finally applied to analyze UA in the samples of human cell cultures exposed to the allergens. The design of this sensor is novel, and the obtained results are interesting. Publication is recommended after addressing the following comments and questions.

1. Please do not use abbreviations in the title. “MWCNT” and “CP” should be replaced with “multi-walled carbon nanotube” and “carbon paste”, respectively.

2. The authors are suggested to describe the sensing principle in the last paragraph of the introduction section.

3. The number in the chemical formulas (e.g., La(OH)3, LaCl3, and K2CO3) should be subscripted.

4. The authors should explain why La(OH)3@MWCNTs/CP shows the highest detection signal among CP, La(OH)3/CP, MWCNTs/CP, and La(OH)3@MWCNTs/CP.

5. The tables look disordered. They should be re-organized.

Reviewer 3 Report

My comments are present in the attached file

Round 2

Reviewer 1 Report

The authors have revised their manuscript. It could be accepted for publication.

Reviewer 3 Report

The paper can be accepted in the present form